# Time to death and its predictors among under-five children on antiretroviral treatment in public hospitals of Addis Ababa, Addis Ababa, Ethiopia, a retrospective follow up study

Enyew Mekonnen[1], Mikias Arega[2], Dawit Misganaw Belay[3]*, Dires Birhanu[4], Tadlo Tesfaw[5], Habtamu Ayele[2], Keralem Anteneh Bishaw[2]

1 Department Midwifery, Saint Peter Specialized Hospital, Federal Minster of Health, Addis Ababa, Ethiopia, 2 Department of Midwifery, College of Medicine and Health Sciences, Debre Markos University, Debre Markos, Ethiopia, 3 Department of Midwifery, College of Health Sciences, Assosa University, Assosa, Ethiopia, 4 Department of Nursing, College of Health Science, Dilla University, Dilla, Ethiopia, 5 Department of Midwifery, Debre Tabor Health Science College, Amhara Regional Health Bureau, Bahir Dar, Ethiopia

* davemisganaw.dm@gmail.com

**Data Availability Statement:** All relevant data are within the paper and its Supporting Information files.

## Abstract

### Background

Child mortality remains a global public health problem, particularly in Sub-Saharan Africa. After initiating ART, the mortality rate among HIV-infected children in Ethiopia was 12–17 deaths per 1000 child-year.

### Objective

To determine the time to death and its predictors among under-five children on antiretroviral treatment in public hospitals of Addis Ababa, Ethiopia, from April 12, 2017, to May 12, 2022.

### Method

An institution-based retrospective follow-up study was conducted among 415 HIV-infected children at selected public hospitals of the Addis Ababa town administration. Computer generated simple random sampling technique was used to select each sampling unit. Data was extracted using a structured data extraction checklist. Data were entered into EPI data 4.2 and analyzed using STATA 14. The child mortality rate was calculated. The Cox proportional hazards regression model was fitted to identify predictor factors. The result of the study was presented using text, tables, graphs, and charts. An adjusted hazard ratio with a 95% confidence interval and a p-value less than 0.05 was used to declare the level of significance.

### Result

A total of 415 (97.42%) of the 426 children on ART were included for analysis. Of these, 41 (9.88%) children were died during the following period. The study participants were followed

**Funding:** The author(s) received no specific funding for this work.

**Competing interests:** The authors have declared that no competing interests exist.

**Abbreviations:** AHR, Adjusted Hazard Ratio; AIDS, Acquired Immune deficiency Syndrome; ART, Anti-Retroviral Treatment; AZT, Zidovudine; CI, Confidence Interval; CHR, Crude Hazard Ratio; EFV, Efavirenz; HAART, Highly Active Anti-Retroviral Therapy; HIV, Human Immunodeficiency Virus; INH, Isoniazid; NVP, Neverapine; PH, Proportional Hazard; 3TC, Lamivudine; PMTCT, Prevention of Mother-to-Child Transmission; SPSS, Statistical Product and Service Solutions; TDF, Tenofovir; WHO, World Health Organization.

for a total of 8237 person- months of risk time. The overall mortality rate was 4.98 (95% CI: 3.67–6.77) per 1000 child-months. The estimated survival after starting ART was 61.42% at 56 months of follow-up. Severe underweight (AHR = 3.19; 95% CI: 1.32–7.71), tuberculosis (AHR = 3.86; CI: 1.76–8.47), low hemoglobin level (AHR = 2.51; CI: 1.02–6.20), and advanced WHO clinical stages at enrolment (AHR = 3.38; CI: 1.08–10.58) were predictors of death among HIV-infected under-five children on ART.

## Conclusion

The incidence of mortality was 4.98 per 1000 child-months. Severe underweight, tuberculosis infection, low hemoglobin level, and advanced WHO clinical stages at enrolment were predictors of death among under-five children on ART.

## Introduction

The child mortality rate is a significant indicator of social and national development since it reflects health equity and access [1]. AIDS is an infection caused by the Human Immunodeficiency Virus (HIV), which predominantly attacks a person's immune system. HIV infection in children under five has rapid disease progression and opportunistic infections due to immune weakness, which leads to higher morbidity and mortality [2–4].

Antiretroviral therapy (ART) is an HIV treatment that includes multiple antiretroviral medications taken throughout life to enhance duration and quality of life by lowering viral load and raising CD4 cell count [5, 6]. The advancement of HIV therapy had a positive impact on disease progression and immunological capacity, resulting in higher survival rates [2]. Ethiopia's government established the first highly active antiretroviral therapy (HAART) guideline in 2003, which was modified in 2021 to scale up the program quickly. According to the 2021 national recommendation, all children should receive HAART [7].

To ensure that simple and effective therapy becomes quickly available and accessible for all under-five children in need, the government, NGO, research partners, health specialists, and civil society vigorously promote 4g = ABC+3TC-LPV/r and 4j = ABC+3TC+DTG for the development of fixed-dose combinations [4].

HIV/AIDS resulted in the death of 328 children daily in 2016, accounting for 1.4% of all under-five deaths globally [8]. The burden is still evident despite a 62% drop in AIDS-related mortality and a 70% drop in new HIV infections since 2000 [9]. In children under the age of five, the condition advances quickly, and if treatment is not initiated, up to 30% of children will die before their first birthday, and up to 50% will die before their second birthday [2]. Without treatment, 35.2% of HIV-infected children in Africa died in their first year, and 52.5% died by the age of 2 (two) [10]. More than half of the deaths (50–68.8%) occurred within 6 (six) months after starting ART [11]. Studies in Africa reported that the incidence of death following ART initiation among children was 18.7% in South Africa [12], Zambia 1.6% [13], Zimbabwe 8.14% [14], Kenya 12.78%, and 2.9% in Cameroon [15].

In Ethiopia, the mortality rate among HIV-infected children in Ethiopia after commencing ART was 12–17 deaths per 1,000 children per year. The length of life of HIV-infected children on ART is mainly affected by clinical factors [16, 17]. A study in Ethiopia's Oromia Liyu Zone found that 5.9% of people died after starting ART [9].

The effects of HIV/AIDS infection happen at a crucial developmental stage when the immune system is immature [18]. On children, wide-ranging factors negatively impact the

quality of care, socialization, health, and emotional and cognitive development of affected children. As a result of their status, they are at a higher risk of mortality, illness, and psychological suffering [19]. The physical well-being of HIV-infected children is likewise heavily reliant on HIV treatment. More than one-third of children do not survive their first birthday if they are not treated [20].

HIV/AIDS has been one of the Ethiopian government's top priorities in the health sector [21]. To achieve this goal, Ethiopia's government has adopted strategies to minimize morbidity and mortality by limiting future disease transmission and boosting access to HIV care, treatment, and support for under-five children living with HIV [22, 23]. Efforts to scale up early baby diagnostic programs and early assessment of HIV-infected infants on ART, regardless of WHO clinical stage or CD4 cell count [24]. More should be done to address the unique survival needs of children in Ethiopia, even though visible efforts to improve the health of HIV-infected children have led to significant reductions in mortality levels among under-five children [5].

Early ART initiation in under five HIV-infected children significantly reduces children's mortality [25]. Even though ART has demonstrated significant clinical importance by achieving the therapy's goal, children continue to die from a variety of preventable causes that could be avoided with appropriate interventions on socioeconomic, demographic, treatment-related, and health factors such as the child's age, CD4 count or CD4 percent at ART initiation, WHO stage, hemoglobin level, and ART adherence, by developing the PMCTC service [12, 26].

ART for HIV-infected children leads to immunological reconstitution through decreasing viral load, increasing CD4 cells, preventing opportunistic infection, and a longer survival time [5, 27]. Even though shreds of evidence regarding is needed regarding the survival status of children on ART and its predictors to maximize the benefit by addressing any modifiable variables, information regarding time to death and its predictors among under-five children after antiretroviral treatment in Ethiopia is limited. In addition, up to the researcher's knowledge, there is a published study in the study setting. Therefore, this study aimed to assess time to death and its predictors among HIV-infected under-five children after initiation of ART treatment at selected public hospitals in Addis Ababa, Ethiopia.

## Methods and materials

### Study design, setting and period

The retrospective follow-up study was conducted to under five children who initiated ART on the period from April 12, 2017, to May 12, 2022 at public hospitals in Addis Ababa, Ethiopia's capital city. Addis Ababa is the headquarters of the African Union. According to the 2007 census, it had a total population of 3,384,569 people, but by 2017, it had about 4 million people, with a population growth rate of 2.9% per year [28]. An estimated 300,000 people are infected with HIV/AIDS. Around 12,000 were youngsters, and 30 to 40 percent of HIV-infected people were under-five children [29].

There are 12 public hospitals in the area. The Addis Ababa health bureau is responsible for managing (Six) of them, including Gandhi Memorial Hospital, Yekatit 12 Hospital Medical College, Zewditu Memorial, Ras-Desta Damtew Memorial, Menelik II Referral, and Tirunesh Beijing General Hospital. Amanuel Hospital, Alert Hospital, St. Peter Specialized Hospital, Millennium Medical College Hospital, and Eka Kotebe Hospital are the five (5) hospitals under the federal Ministry of Health's administration. The other is a teaching hospital called Tikur Anbessa Specialized Hospital, administered by Addis Ababa University. Only eight hospitals provide ART services for under-five children, including Black Lion Hospital, Yekatit 12 Hospitals Medical College, Saint Petrous Specialized Hospital, Alert Hospital, Tirunesh Beijing

General Hospital, St. Paulo's Hospital, Zewditu Memorial Hospital, and Menelik II Referral Hospital, included in the study. The remaining four excluded from the study because they did not provide under five ART services.

## Populations

All under-five children with HIV infection and on ART at the public hospital of Addis Ababa were the source population, and HIV-infected under-five children who started antiretroviral therapy (ART) the period from April 12, 2017, to May 12, 2022, at the selected public hospital of Addis Ababa were the study population.

## Eligibility criteria

All HIV-infected under-five children on antiretroviral therapy (ART) at the selected public hospital of Addis Ababa, from April 12, 2017, to May 12, 2022, were included. Under-five children with incomplete data on the outcome variables like date of treatment initiation, age of the child and pertinent variables were excluded from the study.

## Sample size determination

The study sample size of the study calculated using STATA version 14 statistical software using the formula N = Event(E) /Probability of event (E) by considering the following assumptions; A 95% level of confidence, 5% level of significance, 80% power and 10% assumption of withdrawals [30]. The maximum size of the study was 426 using sex as a predictor(h ratio(2.4), failprob(0.107), wdprob(0.1)) [30].

## Sampling procedure

The study covered all public hospitals that delivered ART to under-five children. The sample size for each hospital was determined using proportional allocation based on cases of under-five children on ART services in the previous five years. A random number was generated based on the identification of each medical chart of children using a computer. Then, four hundred twenty-six (426) children's medical records were selected using a computer-generated random sampling technique. Furthermore, replacing incomplete medical charts was taken into account (**Fig 1**).

## Study variables

**Dependent variable.** Time to death.

**Independent variables. Socio—demographic characteristics:** Age, sex, residence, family status, age of care giver.

**Baseline and follow-up clinical and immunologic information:** TB infection, TB treatment, WHO clinical stages, Cotrimoxazole prophylaxis use, CD4 count, ART adherence, hemoglobin level, chronic diarrhea, drug regimen, and obstetric/birth history of the child.

**Nutritional factors:** Weight for age and developmental milestone.

## Operational definitions

**Time to death:** Time at which the children died during the follow up, after initiation Anti-retroviral treatment [31].

**Follow up time period:** Time from starting of ART up to either the study subjects died or censored.

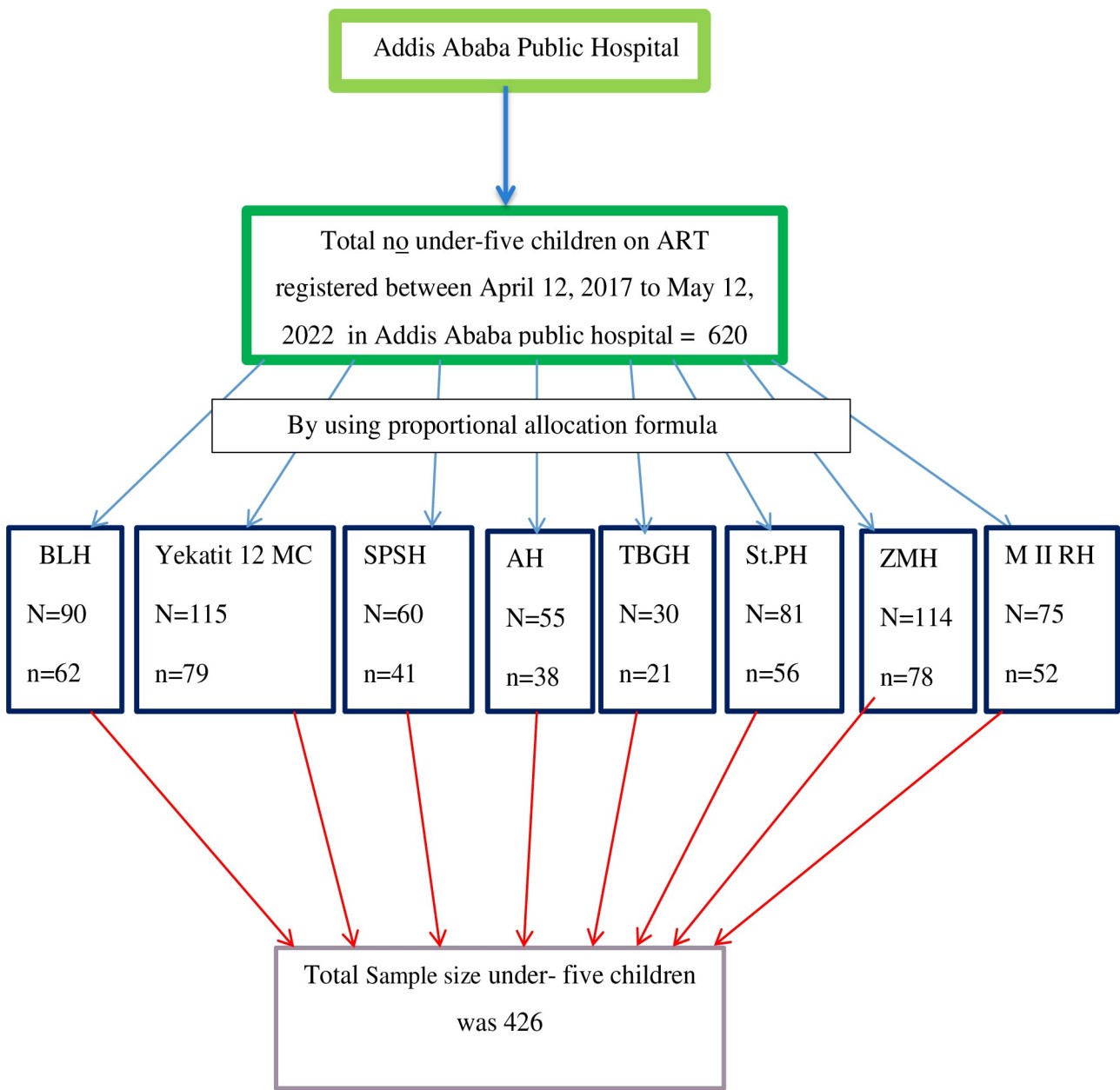

**Fig 1. Schematic representation of sampling procedure for time to death and its predictors among under-five children on antiretroviral treatment in public hospitals of Addis Ababa, Ethiopia 2022.**

**Survival time:** The length of time in months a child was followed from the time the child started ART until death, was lost to follow up, or was still on follow up [32].

**Event:** The occurrence of death from initiation of ART to the end of the study [33].

**Anemia:** Is defined as having hemoglobin level is < 11mg/dl [9].

**CD4 count below threshold:** If CD4 cell count of less than350 cells/mm3 [10].

**Underweight:** For children under the age of five, weight for age Z-score-2 SD was used [34].

**Advanced WHO clinical stages:** Under-five children in WHO clinical stages III and IV during ART enrolment. **Mild WHO clinical stages:** are stage I and II baseline clinical stages of HIV-infected under-five children during antiretroviral therapy (ART) enrolment [35].

**Censored:** When a child is lost-to-follow-up, transferred out, withdrawn from follow-up, and still on follow up(lives longer than the study period) [32].

**Poor adherence to medications:** Less than 90% adherence to ART drugs by children under the age of five in the previous three months was termed poor adherence [36].

**Fair adherence to medications:** Fair adherence was defined as 90–95 percent of children under the age of five who have taken antiretroviral medications in the previous three months [36].

**Good adherence:** adherence level of under-five children to ART medications within the last three months >95%(7) [37].

**Moderate stunting:** under five children having Height/ Age Z-score < −2 SD [37].

**Moderate underweight:** under five children having Weight/Age Z-score < − 2 SD [37].

**Moderate wasting:** under five children having Weight/ Height Z-score < −2 SD [37].

**Severe stunting:** under five children having Height/Age Z-score < −3 SD [37].

**Severe underweight:** under five children having Weight/Age Z-score < −3 SD [37].

**Severe wasting:** under five children having Weigh/ Height Z-score < −3 SD [37].

**Opportunistic infections:** During follow-up periods, an HIV-infected under-five child on ART developed one or more reported histories of opportunistic illnesses [9].

**Developmental milestone:** The child can achieve the four developmental milestones, which are cognitive development, gross motor/fine motor development, language development, and social development, as appropriate for the child's age [35].

**Developmental delay:** is the delay of an age-specific capacity for important developmental milestones and without any deformity [35].

**Developmental regression:** After a time of relatively normal growth, a child loses an already acquired talent or fails to progress beyond a protracted plateau [35].

## Data collection procedure & tools

First, the completeness of children's medical records was checked. Then, a data extraction checklist was prepared. The time from the initiation of ART was the starting point for retrospective follow-up, and the end was the date of death, date of loss to follow-up, or date of transfer out. All HIV-infected children who started ART at the eight hospitals between April 2017 and May 2022 were recruited from the ART follow-up register. Then records were reviewed before the data was collected (both baseline and follow-up records). The death was confirmed by registration follow-up chart, physician recorded, and their medical record number was used to identify them. Eight BSc nurse data collector and Three MSc nurse supervisors were recruited.

## Data quality control

To assure the quality of data, data collectors and supervisors were trained. Data quality was also maintained by using a standard pretested data extraction checklist and ongoing supervision. During data analysis, all collected data were checked for completeness and consistency. Before processing, the data was entered and cleaned.

## Data analysis

Under-five children's death was the event of interest coded as "1" and "0" for censored. Data was entered into Epi-Data 4.6 and analyzed with STATA 14. The child mortality rate was estimated and expressed as 1000 children—months. The Kaplan-Meier curve was used to calculate the median survival time. The Schoenfeld residuals test was used to assess model fitness. The model was fitted, and an overall global test result of 0.6434. Bivariable Cox proportional

hazards regression was fitted, and those ≤0.25 were fitted for multivariable Cox regression hazard model to identify predictors of the outcome variable. The adjusted hazard ratio (AHR) with a 95% confidence interval (CI) and a P 0.05 was used to determine the strength of association and statistical significance. Variance inflation factor (VIF) was used to test for multicollinearity among variables, and there was no evidence of multicollinearity with a variance inflation factor smaller than 4 for all variables.

### Ethical consideration and consent to participants

Ethical approval was sought from the Debre Markos University, College of Health Science Research Ethics Review Committee (Ref. no. HSR/R/C/Ser/PG/Co/175/11/14). A permission letter was secured from each hospital. Moreover, individual consent was not applicable since it is a retrospective study of medical records (record review). The ethics committee waived the requirement for informed consent. Finally, the confidentiality of the information and privacy of study participants was maintained.

## Results

### Socio demographic characteristics of study participants

Based on the inclusion criteria, 415 (97.42%) of the 426 children on ART were included for analysis. More than half of the 226 children (54.5%) were female. More than ninety percent of the children, 390 (94.0%), resides in urban. Almost two-third (61.2%) of the 254 study participants were under the age of 24 months, with the overall mean and standard deviation of the child's age at ART initiation being 23.8 and 14.64 months, respectively. More than two-third of the study's 296 participants (71.3%) were Orthodox Christians. Three hundred and one (72.5%) caregivers were females, and 254 (61.4%) caregivers were between the ages of 20 and 35. More than three-quarters (313) were married, and less than half of caregivers attended primary school (**Table 1**).

### Baseline clinical characteristics of children under five years on ART

The majority, 388 (93.5%) of children, had appropriate developmental status, and more than half, 224(54.0%), children started INH prophylaxis. Two hundred forty-three (58.6%) of children started ART in the mild WHO clinical disease stage of HIV (II or I), and 240(57.8%) of children had less than 11 grams per deciliter (g/dl) of hemoglobin. Four hundred (96.9%) of children were using Cotrimoxazole, and less than a quarter of 60(14.5%) children had tuberculosis. Almost two third children, 271 (65.3%), had good ART adherence. Three hundred thirty-three (80.2%) of HIV-infected under-five children were born from mothers who had PMTCT service initiated during pregnancy. More than three-fourth, 341(82.2%) of children had chronic diarrhea, and 388 (93.5%) of children had above the threshold CD4 count (**Table 2**).

Regarding, opportunistic infection, 152(36.6%) had not developed an opportunistic infection. More than one-fourth (34.22%) of children had herpes zoster, and less than a quarter (12.77%) of children had bacterial pneumonia (**Fig 2**). More than half, 209 (50.36%) of children have taken a drug of type 4g = ABC+3TC-LPV/r ART regimen followed by 197(47.47%) of children who have taken a drug of type 4j = ABC+3TC+DTG (**Fig 3**).

### Nutritional status of children on ART

This study found that 141 (34.4%) of children were severely stunted, whereas 35 (8.4%) were moderately stunted. More than one-fourth of the children, 123(29.6%), were very

Table 1. Socio-demographic characteristics of under-five children on antiretroviral treatment in public hospitals of Addis Ababa, Ethiopia 2022. (n = 415).

| Variables | Categories | Frequency | Percent |
|---|---|---|---|
| Child's a sex | Male | 189 | 45.5 |
| | Female | 226 | 54.5 |
| age of child | Less than 24 months | 254 | 61.2 |
| | 25–59 months | 161 | 38.8 |
| Residence | Urban | 390 | 94.0 |
| | Rural | 25 | 6.0 |
| Religion | Orthodox | 296 | 71.3 |
| | Protestant | 78 | 18.8 |
| | Catholic | 31 | 7.5 |
| | Other | 10 | 2.4 |
| Sex of care giver | Male | 114 | 27.5 |
| | Female | 301 | 72.5 |
| age of care givers in years | <20 | 6 | 1.4 |
| | 20–35 | 254 | 61.4 |
| | >35 | 155 | 37.3 |
| Marital status of care giver | Married | 313 | 75.4 |
| | Unmarried | 102 | 24.6 |
| Primary care giver | Parents | 385 | 92.8 |
| | Relatives | 23 | 5.5 |
| | Guardians | 7 | 1.7 |
| Educational level of the care giver | No education | 69 | 16.6 |
| | Primary | 180 | 43.4 |
| | Secondary | 132 | 31.8 |
| | Tertiary and above | 34 | 8.2 |
| Family HIV status currently | Both alive | 352 | 84.8 |
| | Father died | 30 | 7.2 |
| | Mother died | 17 | 4.1 |
| | Both died | 16 | 3.9 |

underweight, and 45(10.8%) were moderately underweight. In addition, 79 (19.0%) and 48 (11.1%) of the children were severely and moderately wasting, respectively (**Table 3**).

## Kaplan Meier analysis

The study reported that 415 children on ART followed up for a minimum of 1 month and a maximum of 56 months, with a median follow-up time of 17 with an interquartile range of (10–29) months. The survival status was estimated by the Kaplan-Meier estimation method. This finding showed that overall estimated survival after starting ART was 61.42% (95%CI: 19.93–86.31) at 56 months of follow-up (**Fig 4**). The Kaplan-Meier curve also revealed that the survival probability of children declined over time. The highest mortality rate occurred in the first 12 months (**Fig 4**).

The study reported the incidence of death was 4.98 /1000 person months observations (95% CI: 3.67–6.77). During the follow-up period, 41 were died and the remaining 374 (90.12%) were censored (**Fig 5**).

The Kaplan-Meier survival curve showed that children with hemoglobin level > 11 g/L had a higher survival probability compared to children with a hemoglobin level <11 g/L (**Fig 6**). Compared to children with CD4 counts above the threshold, those with CD4 counts below it

**Table 2. Baseline clinical characteristics of under-five children on antiretroviral treatment in public hospitals of Addis Ababa, Ethiopia 2022.** (n = 415).

| Variables | Categories | Outcome. | | Total, n (%) |
|---|---|---|---|---|
| | | Died, n (%) | Censored, n (%) | |
| Developmental status | Appropriate for age | 37(90.2) | 351(93.9) | 388(93.5) |
| | Delayed | 4(9.8) | 17(4.5) | 21(5.1) |
| | Regressed | - | 6(1.6) | 6(1.4) |
| INH prophylaxis | Yes | 15(36.6) | 209(55.9) | 224(54.0) |
| | No | 26(63.4) | 165(44.1) | 191(46.0) |
| WHO Clinical stage | Mild | 9(22.0) | 234(62.6) | 243(58.6) |
| | Advanced | 32(78.0) | 140(37.4) | 172(41.4) |
| Hemoglobin | <11 | 34(82.9) | 206(55.1) | 240(57.8) |
| | >11 | 7(17.1) | 168(44.9) | 175(42.2) |
| Cotrimoxazole use | Yes | 39(95.1) | 363(97.1) | 402(96.9) |
| | No | 2(4.9) | 11(2.9) | 13(3.1) |
| Immunization Status | Immunized | 23(56.1) | 227(60.7) | 250(60.2) |
| | Not immunized at all | 18(43.9) | 147(39.3) | 165(39.8) |
| Tuberculosis | Yes | 19(46.3) | 41(11.0) | 60(14.5) |
| | No | 22(53.7) | 333(89.0) | 355(85.5) |
| Comorbidities other than TB | Yes | 33(80.5) | 228(61.0) | 261(62.9) |
| | No | 8(19.5) | 146(39.0) | 154(37.1) |
| ART adherence | Good | 11(26.8) | 260(69.5) | 271(65.3) |
| | Fair | 26(63.4) | 111(29.7) | 137(33.0) |
| | Poor | 4(9.8) | 3(0.8) | 7(1.7) |
| Maternal history of PMTCT | Yes | 30(73.2) | 303(81.0) | 333(80.2) |
| | No | 11(26.8) | 71(19.0) | 82(19.8) |
| Drugs side effect | Yes | 36(87.8) | 273(73.0) | 309(74.5) |
| | No | 5(12.2) | 101(27.0) | 106(25.5) |
| Chronic Diarrhea | Yes | 37(90.2) | 304(81.3) | 341(82.2) |
| | No | 4(9.8) | 70(18.7) | 74(17.8) |
| Mode of delivery | Cesarean section | 1(2.4) | 67(17.9) | 68(16.4) |
| | SVD | 40(97.6) | 307(82.1) | 347(83.6) |
| Place of delivery | Health facility | 33(80.5) | 335(89.6) | 368(88.7) |
| | Home | 8(19.5) | 39(10.4) | 47(11.3) |
| CD4 count | below threshold | 6(14.6) | 21(5.6) | 27(6.5) |
| | above the threshold | 35(85.4) | 353(94.4) | 388(93.5) |

had a greater chance of surviving at 2 and 3 months. But starting from 4 months onwards, children with a CD4 level above the threshold had a better chance of surviving than those with a CD4 count below the threshold (**Fig 7**). This study also showed that children who used Cotrimoxazole for the first two months had a higher survival probability compared to those who did not. However, their chance of survival decreased from the third to the 34th month (**Fig 8**).

## Predictors of time to death of children on ART

Bivariable Cox proportional hazard regression revealed that sex, age of a child, sex of caregiver, stunting, underweight, INH prophylaxis, WHO stage, hemoglobin level, tuberculosis, opportunistic infection, maternal history of PMTCT, drug side effects, place of delivery, and CD4 count variables were associated with time to death of children on ART follow up at a P-value less than 0.25. But underweight, having tuberculosis, low hemoglobin level, and WHO clinical

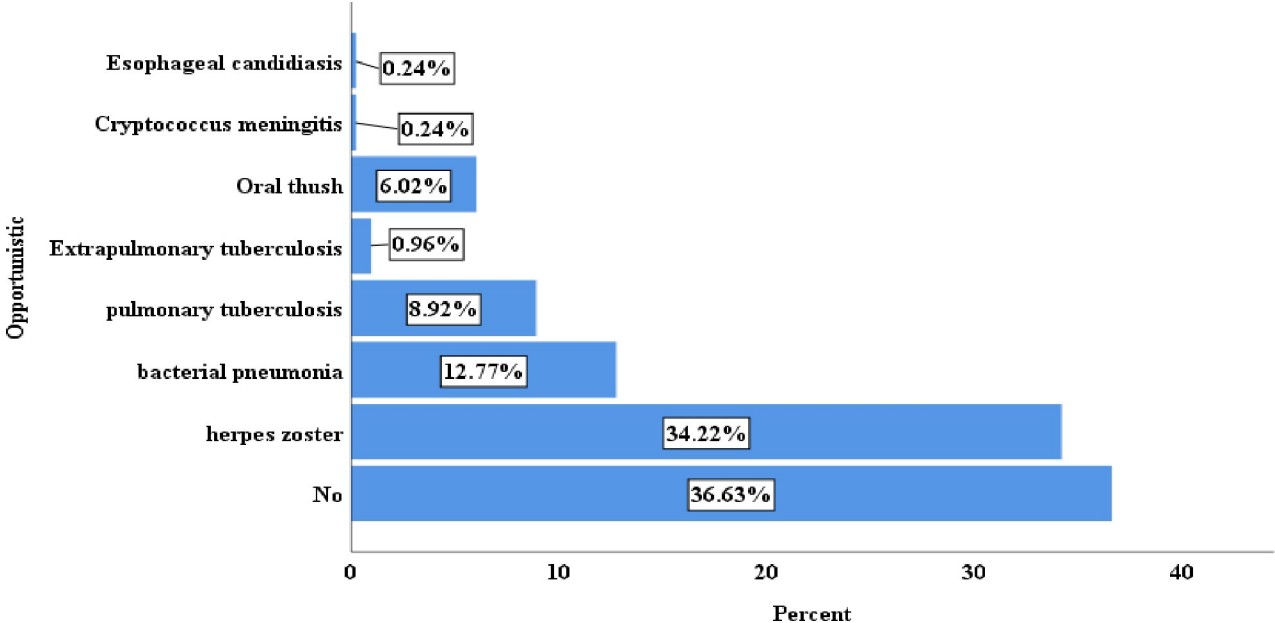

**Fig 2. Baseline opportunistic infection among under-five children on antiretroviral treatment in public hospitals of Addis Ababa, Ethiopia, 2022.**

stage remained statistically significant with time to death among children on ART in multivariable Cox proportional hazards regression (**Table 4**).

The study found that severely underweight children on ART are more like to die (AHR = 3.19) than children with normal weight (AHR = 3.19; 95%CI: 1.32–7.71). Children with tuberculosis were 3.86 times more likely to die than their counterparts (AHR = 3.86; CI: 1.76–8.47). And also, children with low hemoglobin levels were 2.51 times more likely to die than children with hemoglobin levels above the threshold (AHR = 2.51; CI: 1.02–6.20). Furthermore, the risk of death among under-five children with advanced WHO clinical stages at

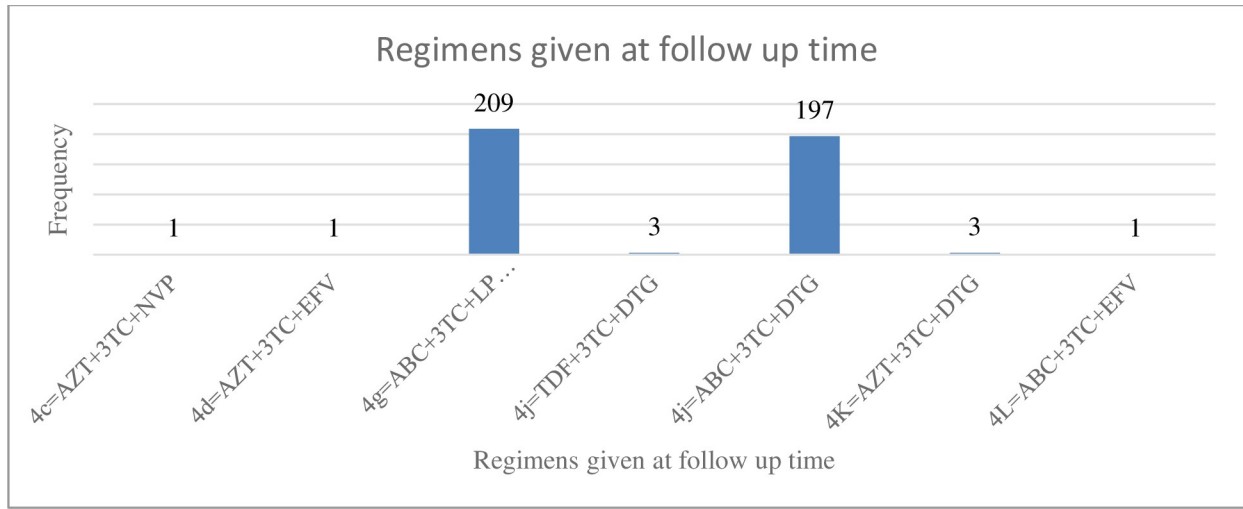

**Fig 3. ART regimen given at baseline among under-five children on antiretroviral treatment in public hospitals of Addis Ababa, Ethiopia, 2022.**

**Table 3. Nutritional status of under-five children on antiretroviral treatment in public hospitals of Addis Ababa, Ethiopia, 2022.** (n = 415).

| Variables | Categories | Outcome | | Total, n (%) |
|---|---|---|---|---|
| | | Died, n (%) | Censored, n (%) | |
| Stunting | Normal | 18(43.9) | 221(59.1) | 239(57.6) |
| | Moderately stunted | 3(7.3) | 32(8.6) | 35(8.4) |
| | Severely stunted | 20(48.8) | 121(32.4) | 141(34.0) |
| Underweight | Normal | 18(43.9) | 229(61.2) | 247(59.5) |
| | Moderately underweight | 5(12.2) | 40(10.7) | 45(10.8) |
| | Severely underweight | 18(43.9) | 105(28.1) | 123(29.6) |
| Wasting | Normal | 28(68.3) | 260(69.5) | 288(69.4) |
| | Moderately wasting | 5(12.2) | 43(11.5) | 48(11.6) |
| | Severely wasting | 8(19.5) | 71(19.0) | 79(19.0) |

enrolment to ART was 3.38 times higher than children with a mild stage (AHR = 3.38; CI:1.08–10.58) (**Table 4**).

A log-rank test was done to check the existence of significant differences in the survival status among categories. Variables like CD4 count, opportunistic infection, INH prophylaxis, and severely stunted had the statistically significant result of the log-rank test with (P-Value <0.05) (**Table 5**).

## Test of proportional hazard assumption

The result showed (p-value > 0.05) satisfy the PH assumptions (**Table 6**).

## Discussions

The study aimed to assess time to death and its predictors among under-five children on ART. The study reported that the cumulative incidence rate after starting ART was 4.98 /1000-person month observations with a 95% CI: (3.67–6.77). A study in the Oromiya Liyu zone of Amhara region, Ethiopia is higher than this study (5.9 deaths per 100 child months [9]. This is because of the difference in study participants, which was conducted among TB/AIDS co-infected children and resulted in higher morality than this study. But this result is higher than

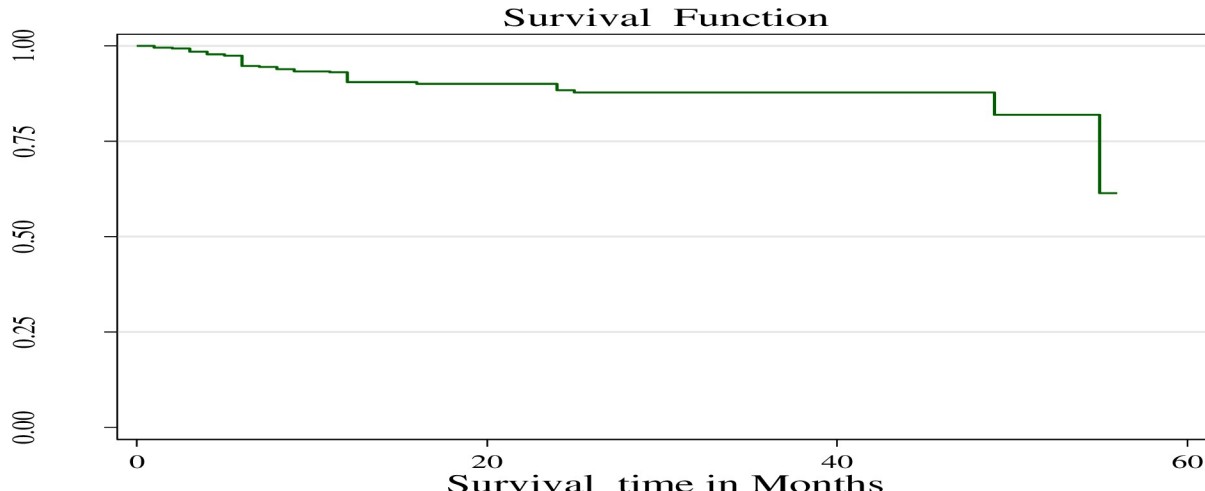

**Fig 4. The Kaplan-Meier survival function among under-five children on ART in public hospitals of Addis Ababa, Ethiopia, 2022.**

## Overall outcome

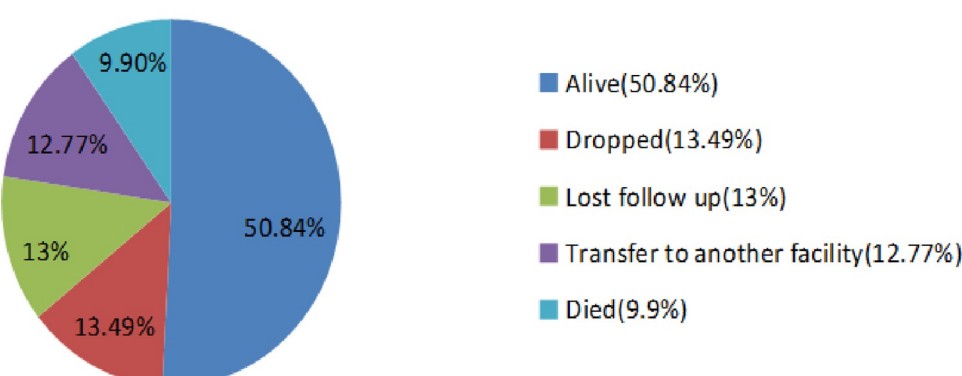

**Fig 5. Over all outcomes of under five children on antiretroviral treatment in public hospitals of Addis Ababa, Ethiopia, 2022.**

a study in Malawi, in which the incidence rate was 16.6 per 1000 children years observation [38]. Differences in the quality of ART services between countries may contribute to this discrepancy. And also, this finding is higher than a study done in West Amhara referral hospitals which reported that the overall incidence rate of death was 2.87 deaths per 1000 child months [39]. The difference might be due to the difference in the study setting.

The study found that the overall estimated survival time of children after starting ART was 61.42% at 56 months of follow-up and had a median follow-up time of 17 months. This finding is lower than a study in West Amhara referral hospitals (85%) [39] and the Oromiya Liyu zone (87%) [9]. This might be explained in terms of differences in clinical characteristics of study participants like WHO clinical stage and CD4 level. A difference in the stage of HIV/AIDS at the beginning of ART could be another explanation. While some study participants did not start ART at the WHO clinical stage I&II. Children who began receiving ART earlier had a longer survival time compared counterparts. This finding is also lower than a study done in Europe which showed that the survival rate after starting ART was 97.6 [40]. The discrepancy may be because developed countries like Europe offer high-quality ART care, resulting in a higher survival probability for children compared to developing countries like Ethiopia.

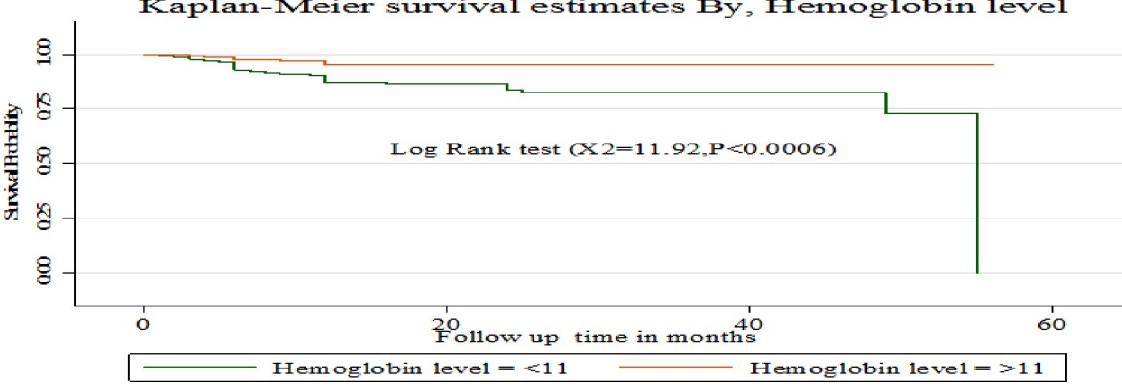

**Fig 6. Kaplan-Meier survival estimates based on hemoglobin level among under- five children on ART treatment in public hospitals of Addis Ababa, Ethiopia, 2022.**

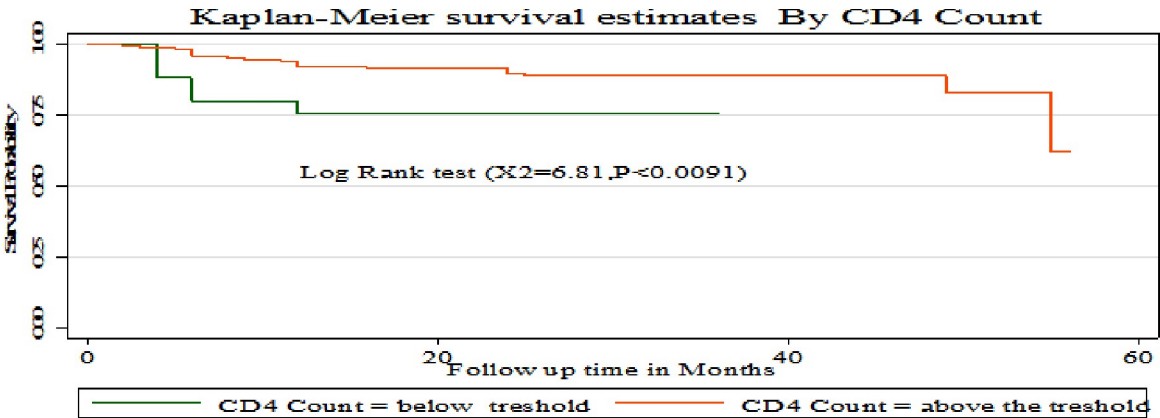

**Fig 7. Kaplan Meier survival estimates based on CD4 count among under- five children on antiretroviral treatment (ART) in public hospitals of Addis Ababa, Ethiopia, 2022.**

This study revealed that the risk of death among under-five children with severe underweight was 3.19 times higher than those children who had normal weight. Studies in Swaziland and Kenya supported this evidence [41, 42]. In addition, this finding is consistent with studies in Ethiopia [43, 44]. This may be due to severely underweight children may not respond well to treatment because their immunity is weakened due to inadequate nutrient delivery to their cells.

This study also revealed children with tuberculosis had a 3.86 times higher risk of death than children without the disease. This finding was consistent with the study conducted in Tanzania [45] and Jinka Hospital of Ethiopia [46]. This could be due to host reactions to M. tuberculosis that promote HIV replication, speeding up the natural progression of the virus and further suppressing immunity [47]. Another explanation for the increased risk of death among tuberculosis-infected children is the reduced absorption of ART medications, which reduces their efficiency secondary to anti-tuberculosis drugs [48].

In this study, the hazard of death among children on ART with advanced stages (stages III and IV) enrolment was 3.38 times higher than with mild stages (stage I and II) at ART

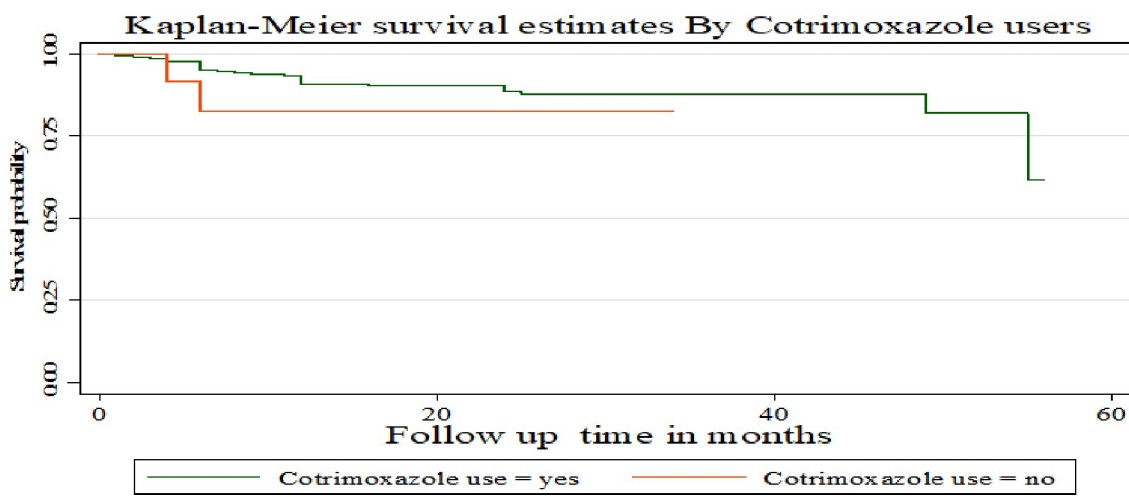

**Fig 8. Kaplan-Meier survival estimates based on cotrimoxazole use under- five children on antiretroviral treatment (ART) in public hospitals of Addis Ababa, Ethiopia, 2022.**

**Table 4. Cox -regression analysis of predictors of death among under-five children on antiretroviral treatment in public hospitals of Addis Ababa, Ethiopia 2022.**

| Variables | Categories | Outcome | | CHR (95% CI) | AHR (95.% CI) | P-value |
|---|---|---|---|---|---|---|
| | | Event (%) | Censored (%) | | | |
| Sex | Male | 23(12.2) | 166(87.8) | 1.00 | 1.00 | |
| | Female | 18(8.0) | 208(92.0) | 0.69(0.37, 1.28) | 0.81(0.40,1.65) | 0.564 |
| Age of child | < 24 months | 23(9.1) | 231(90.9) | 0.64(0.34, 1.21) | 0.54(0.26,1.11) | 0.093 |
| | 25–59 month | 18(11.2) | 143(88.8) | 1.00 | 1.00 | |
| Sex of caregiver | Male | 8(7.0) | 106(93.0) | 0.58(0.27, 1.26) | 0.51(0.22,1.17) | 0.11 |
| | Female | 33(11.0) | 268(89.0) | 1.00 | 1.00 | |
| Stunting | Normal | 18(7.5) | 221(92.5) | 1.00 | 1.00 | |
| | Moderately | 3(8.6) | 32(91.4) | 1.07(0.31, 3.65) | 0.99 (0.25,3.85) | 0.98 |
| | Severely | 20(14.2) | 121(85.8) | 2.28(1.19, 4.35) | 1.32 (0.25, 3.04) | 0.52 |
| Under Weight | Normal | 18(7.3) | 229(92.7) | 1.00 | 1.00 | |
| | Moderately | 5(11.1) | 40(88.9) | 1.65(0.61, 4.47) | 2.24(0.73,6.93) | 0.16 |
| | Severely | 18(14.6) | 105(85.4) | 2.47(1.27, 4.80) | 3.19(1.32,7.71) | **0.01*** |
| INH prophylaxis | Yes | 15(6.7) | 209(93.3) | 0.48(0.25, 0.91) | 0.54(0.26, 1.12) | 0.9 |
| | No | 26(13.6) | 165(86.4) | 1.00 | 1.00 | |
| WHO stage | Mild | 9(3.7) | 234(96.3) | 1.00 | 1.00 | |
| | Advanced | 32(18.6) | 140(81.4) | 6.08(2.80,13.21) | 3.38(1.08, 10.58) | **0.037*** |
| Hemoglobin | <11g/dl | 34(14.2) | 206(85.8) | 3.80(1.68, 8.62) | 2.51(1.02,6.20) | **0.046*** |
| | >11g/dl | 7(4.0) | 168(96.0) | 1.0* | 1.00 | |
| Tuberculosis | Yes | 19(31.7) | 41(68.3) | 5.56(2.99,10.34) | 3.86(1.76, 8.47) | **0.001*** |
| | No | 22(6.2) | 333(93.8) | 1.00 | 1.00 | |
| Opportunistic infection | Yes | 33(12.5) | 230(87.5) | 2.52(1.16, 5.47) | 0.89(0.30, 2.68) | 0.84 |
| | No | 8(5.3) | 144(94.7) | 1.00 | 1.00 | |
| Maternal history of PMTCT | Yes | 30(9.0) | 303(91.0) | 0.62(0.31, 1.24) | 1.73(0.76, 3.94) | 0.20 |
| | No | 11(13.4) | 71(86.6) | 1.00 | 1.00 | |
| Drugs side effect | Yes | 36(11.7) | 273(88.3) | 2.42(0.95, 6.18) | 0.90(0.27, 2.92) | 0.85 |
| | No | 5(4.7) | 101(95.3) | 1.00 | 1.00 | |
| Place of delivery | Health facility | 33(9.0) | 335(91.0) | 1.00 | 1.00 | |
| | Home | 8(17.0) | 39(83.0) | 1.98(0.91, 4.30) | 1.35(0.55, 3.32) | 0.508 |
| CD4 | Below threshold | 6(22.2) | 21(77.8) | 3.00(1.25, 7.17) | 1.71(0.58,5.01) | 0.329 |
| | Above the threshold | 35(9.0) | 353(91.0) | 1.00 | 1.00 | |

**Key:** 1.00 = Reference; * = statistically significant at a p < 0.05

enrolment. Studies Gondar of Ethiopia [49] and Cameroon [15] supported this finding. This could be because children with advanced HIV/AIDS have compromised immune systems and are more vulnerable to opportunistic infections, which can lead to mortality. This reflects the fact that severe immunodeficiency increases the risk of death.

Finally, the study revealed that the risk of death among under-five children with low hemoglobin was 2.51 times higher than compared to counterparts. Studies in Gondar and Addis Ababa, Ethiopia [16, 50] supported this evidence. This could be because low hemoglobin speeds up the progression of HIV infection, which increased the risk of death. Combined effects of anemia and some drugs worsen anemia, and lead to more comorbid conditions and death.

The finding has significant clinical implications for reducing the death rate of under-five children by enhancing nutritional status, increasing hemoglobin level, preventing TB, and preventing advanced HIV infection in under-five children. Additionally, this finding is important

**Table 5. Log-rank test for equality of survivor functions among under- five children on ART treatment in public hospitals of Addis Ababa, Ethiopia, 2022.**

| Variables | Categories | P-value | Log-rank test ($x^2$) |
|---|---|---|---|
| sex of child | Male | 0.2328 | 1.42 |
| | Female | | |
| age of child | < 24 months | 0.1658 | 1.92 |
| | 25–59 month | | |
| sex of care giver | Male | 0.1616 | 1.96 |
| | Female | | |
| Underweight | Normal | 0.0210 | 7.72 |
| | Moderately underweight | | |
| | Severely underweight | | |
| Stunting | Normal | 0.0291 | 7.07 |
| | Moderately stunted | | |
| | Severely stunted | | |
| INH prophylaxis | Yes | 0.0199 | 5.42 |
| | No | | |
| WHO clinical Stage | Mild | 0.0000 | 27.40 |
| | Advanced | | |
| hemoglobin level | <11g/dl | 0.0006 | 11.92 |
| | >11g/dl | | |
| Tuberculosis | Yes | 0.0000 | 37.73 |
| | No | _ | _ |
| opportunistic infection | Yes | 0.0145 | 5.98 |
| | No | _ | _ |
| maternal history of PMTCT | Yes | 0.1690 | 1.89 |
| | No | | |
| Drug side effect | Yes | 0.0542 | 3.71 |
| | No | _ | _ |
| place of delivery | Health facility | 0.0757 | 3.16 |
| | Home | | |
| CD4 count | Below threshold | 0.0091 | 6.81 |
| | Above the threshold | _ | _ |

for policymakers in developing integrated therapeutic interventions and strategies to prevent tuberculosis, advanced stage of HIV/AIDS, underweight, and low hemoglobin level.

## Limitation of the study

Because the study was based on secondary data, some important variables, such as sociodemographic and clinical characteristics, missed due to poor documentation and thus excluded from the study. Additionally, those study participants whose medical chat was not available were not included, in the study, which may undermine the study's findings.

## Conclusion and recommendations

The incidence of mortality was 4.98 per 1000 child-months. Severe underweight, tuberculosis infection, low hemoglobin level, and advanced WHO clinical stages at enrolment were predictors of death among under-five children on ART. Therefore, intervention to reduce the mortality of HIV-infected children should be considered. And also, responsible bodies should

**Table 6. Schoenfeld's residuals test result among under-five children on antiretroviral treatment in public hospitals of Addis Ababa, Ethiopia 2022.**

| Variable | Rho | chi2 | Df | prob>chi2 |
|---|---|---|---|---|
| Sex of child | 0.011 | 0.01 | 1 | 0.944 |
| Age of child | -0.070 | 0.18 | 1 | 0.668 |
| Sex of care giver | -0.158 | 1.10 | 1 | 0.295 |
| Underweight | -0.156 | 1.26 | 1 | 0.261 |
| Stunting | 0.120 | 0.62 | 1 | 0.430 |
| INH prophylaxis | -0.055 | 0.13 | 1 | 0.717 |
| WHO clinical Stage | 0.011 | 0.01 | 1 | 0.932 |
| Hemoglobin level | -0.210 | 1.71 | 1 | 0.191 |
| Tuberculosis | -0.024 | 0.03 | 1 | 0.865 |
| Opportunistic infection | -0.042 | 0.08 | 1 | 0.776 |
| Maternal history of PMTCT | -0.166 | 0.88 | 1 | 0.347 |
| Drug side effect | -0.185 | 1.16 | 1 | 0.282 |
| Place of delivery | 0.162 | 0.89 | 1 | 0.345 |
| CD4 count | -0.003 | 0.00 | 1 | 0.987 |
| Global test | 11.54 | | 14 | **0.6434** |

implement strategies to improve children's weight, prevent tuberculosis infection and maintain a high level of hemoglobin.

## Supporting information

**S1 File.**
(SAV)

## Acknowledgments

We would like to acknowledge Debre Markos University. We also greatly acknowledge Saint Peter specialized hospital and Addis Ababa public hospital workers especially ART clinic staffs.

## Author Contributions

**Conceptualization:** Enyew Mekonnen, Mikias Arega, Dawit Misganaw Belay, Dires Birhanu, Tadlo Tesfaw, Habtamu Ayele.

**Data curation:** Enyew Mekonnen, Mikias Arega, Dawit Misganaw Belay.

**Formal analysis:** Enyew Mekonnen, Mikias Arega, Dawit Misganaw Belay, Dires Birhanu, Tadlo Tesfaw, Keralem Anteneh Bishaw.

**Funding acquisition:** Enyew Mekonnen, Habtamu Ayele.

**Investigation:** Enyew Mekonnen, Mikias Arega, Dawit Misganaw Belay, Dires Birhanu, Tadlo Tesfaw, Keralem Anteneh Bishaw.

**Methodology:** Enyew Mekonnen, Mikias Arega, Dawit Misganaw Belay, Dires Birhanu, Tadlo Tesfaw, Keralem Anteneh Bishaw.

**Project administration:** Enyew Mekonnen.

**Resources:** Enyew Mekonnen.

**Software:** Enyew Mekonnen, Dawit Misganaw Belay.

**Supervision:** Enyew Mekonnen, Mikias Arega, Dawit Misganaw Belay.

**Validation:** Enyew Mekonnen, Mikias Arega, Dawit Misganaw Belay, Dires Birhanu, Tadlo Tesfaw, Habtamu Ayele, Keralem Anteneh Bishaw.

**Visualization:** Enyew Mekonnen, Mikias Arega, Dawit Misganaw Belay, Dires Birhanu, Tadlo Tesfaw, Habtamu Ayele, Keralem Anteneh Bishaw.

**Writing – original draft:** Enyew Mekonnen, Dawit Misganaw Belay.

**Writing – review & editing:** Enyew Mekonnen, Mikias Arega, Dawit Misganaw Belay, Dires Birhanu, Tadlo Tesfaw, Habtamu Ayele, Keralem Anteneh Bishaw.

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
