## [Decision Letter · Decision Letter 0]

2 Mar 2023

PONE-D-22-31887Time To Death and Its Predictors Among Under-Five Children on Antiretroviral Treatment in Public Hospitals of Addis Abbaba, Addis Abbaba, Ethiopia, A Retrospective Chohort StudyPLOS ONE

Dear Dr. Belay,

Thank you for submitting your manuscript to PLOS ONE. After careful consideration, we feel that it has merit but does not fully meet PLOS ONE’s publication criteria as it currently stands. Therefore, we invite you to submit a revised version of the manuscript that addresses the points raised during the review process. Please submit your revised manuscript by Apr 16 2023 11:59PM. If you will need more time than this to complete your revisions, please reply to this message or contact the journal office at plosone@plos.org. Please include the following items when submitting your revised manuscript:A rebuttal letter that responds to each point raised by the academic editor and reviewer(s). You should upload this letter as a separate file labeled 'Response to Reviewers'.A marked-up copy of your manuscript that highlights changes made to the original version. You should upload this as a separate file labeled 'Revised Manuscript with Track Changes'.An unmarked version of your revised paper without tracked changes. You should upload this as a separate file labeled 'Manuscript'.

We look forward to receiving your revised manuscript.

Kind regards,

Daniel Bekele Ketema, MPH

Academic Editor

PLOS ONE

Journal Requirements:

Reviewers' comments:

Reviewer's Responses to Questions

**Comments to the Author**

1. Is the manuscript technically sound, and do the data support the conclusions?

Reviewer #1: Yes

Reviewer #2: Yes

2. Has the statistical analysis been performed appropriately and rigorously? 

Reviewer #1: Yes

Reviewer #2: Yes

3. Have the authors made all data underlying the findings in their manuscript fully available?

Reviewer #1: Yes

Reviewer #2: Yes

4. Is the manuscript presented in an intelligible fashion and written in standard English?

Reviewer #1: No

Reviewer #2: No

5. Review Comments to the Author

Reviewer #1: The manuscript technically sound, and do the data support the conclusion

The analysis was done with appropirate statistical method.They should improve the quality of the English throughout the manuscript,since it needs proofreading for grammatical and typographical errors.

Reviewer #2: Title: Time To Death and Its Predictors Among Under-Five Children on Antiretroviral Treatment in Public Hospitals of Addis Abbaba, Addis Abbaba, Ethiopia, A Retrospective Chohort Study

Comment #1: The manuscript is within the scope of the journal. However, although the paper is not novel, the information generated may help program planners. By using the proper statistical notation, the author might enhance how the data is presented. For ease of interpretation, the author could decide to convert the data into figures. The author might also need to make more edits to the content to improve the language and other terminologies.

Comment#2: The authors did a fantastic job of putting the manuscript together, but it was tough for me to read line by line because they didn't number the pages or the sentences. This is typical manuscript procedure.

6. PLOS authors have the option to publish the peer review history of their article (what does this mean?). If published, this will include your full peer review and any attached files.

Reviewer #1: No

Reviewer #2: **Yes: **Addis Birhanu

---

## [Author Response · Author response to Decision Letter 0]

13 Mar 2023

Dear editor and reviewers, I hope now the manuscript is clear and more acceptable than the previous one. We have tried to present the manuscript in proper manner according to your comments. Finally, I would like to extend our deepest gratitude for your endeavor for the improvement of this manuscript.

Both Reviewers and academic editor assigned for me were excellent. Thank you Daniel Bekele Ketema, Addis Birhanu and the rest one for your smart suggestions and comments.

Respectfully, 

Dawit Misganaw (Corresponding author)

---

## [Decision Letter · Decision Letter 1]

24 Apr 2023

PONE-D-22-31887R1Time to death and its predictors among under-five children on antiretroviral treatment in public hospitals of Addis Ababa, Addis Ababa, Ethiopia, A Retrospective Cohort StudyPLOS ONE

Dear Dr. Belay,

Thank you for submitting your manuscript to PLOS ONE. After careful consideration, we feel that it has merit but does not fully meet PLOS ONE’s publication criteria as it currently stands. Therefore, we invite you to submit a revised version of the manuscript that addresses the points raised during the review process.

The manuscript should benefit from experienced language editors. The authors should mention who made the language edition in the manuscript during submissions.In addition, the authors should justify the novelty of this paper and how it contributes to the existing evidence.=============================================================================================================================

We look forward to receiving your revised manuscript.

Kind regards,

Daniel Bekele Ketema

Academic Editor

PLOS ONE

Journal Requirements:

Reviewers' comments:

Reviewer's Responses to Questions

**Comments to the Author**

1. If the authors have adequately addressed your comments raised in a previous round of review and you feel that this manuscript is now acceptable for publication, you may indicate that here to bypass the “Comments to the Author” section, enter your conflict of interest statement in the “Confidential to Editor” section, and submit your "Accept" recommendation.

Reviewer #1: All comments have been addressed

Reviewer #2: (No Response)

2. Is the manuscript technically sound, and do the data support the conclusions?

Reviewer #1: Yes

Reviewer #2: Yes

3. Has the statistical analysis been performed appropriately and rigorously? 

Reviewer #1: Yes

Reviewer #2: Yes

4. Have the authors made all data underlying the findings in their manuscript fully available?

Reviewer #1: No

Reviewer #2: Yes

5. Is the manuscript presented in an intelligible fashion and written in standard English?

Reviewer #1: Yes

Reviewer #2: No

6. Review Comments to the Author

Reviewer #1: Comments to the Author

Almost all my comments are addressed to my satisfaction - thanks to the authors.

There are still some grammatical mistakes. I would advise authors to carefully read the manuscript and improve the quality of the writing throughout your manuscript.

For example, on page 8 lines 214-215, the statement is not clear.

Reviewer #2: Comments to the editor and authors

The results of this study are not novel, but they add to the body of knowledge about the Time to Death and its Predictors among Children Under Five and provide local evidence. The study is quite straightforward and uncomplicated, but there are a number of ways that the content might be improved to be more appropriate for Plos One Journal readers. The language could require some editing and refinement.

Please find the following feedback and advice for your consideration in order to further improve your manuscript:

7. PLOS authors have the option to publish the peer review history of their article (what does this mean?). If published, this will include your full peer review and any attached files.

Reviewer #1: No

Reviewer #2: **Yes: **Addis Birhanu

---

## [Author Response · Author response to Decision Letter 1]

14 May 2023

Comments and recommendations have been modified and heighted on the main document. Questions and some points which need clarifications are listed under "response to reviewers". 

Dear reviewers, we hope now the manuscript is clear and more acceptable than the previous one. We have tried to present the manuscript in proper manner according to your comments. 

We would like to extend our deepest gratitude for your endeavor for the improvement of this manuscript.

Finally, both our Reviewers and the academic editor assigned for us were excellent. Thank you all for your smart suggestions and comments.

---

## [Editor Report · Decision Letter 2]

29 May 2023

PONE-D-22-31887R2Time to death and its predictors among under-five children on antiretroviral treatment in public hospitals of Addis Ababa, Addis Ababa, Ethiopia, A Retrospective Follow up StudyPLOS ONE

Dear Dr. Belay,

Thank you for submitting your manuscript to PLOS ONE. After careful consideration, we feel that it has merit but does not fully meet PLOS ONE’s publication criteria as it currently stands. Therefore, we invite you to submit a revised version of the manuscript that addresses the points raised during the review process.

ACADEMIC EDITOR'S COMMENTS

The manuscript was well improved based on the reviewers comments. Based on my assessment, the  manuscript still needs some methodological improvement. As a result, I gave the following specific feedbacks to be addressed prior to formal acceptance. 

**General comment**

There were missing, discordant, and incomplete statements throughout the document. As such, it still requires careful language editing before publication.

**Specific methodological comments **

**   Abstract**

The authors should describe the name of the hospitals included in the studyThe measure of association with its level of precision (aHR with 95% CI) line #29-30: good to explain why these statistical computations were performed (Long rank test for comparing survival curves across different baseline variables; do the same for the remaining.)Study period (April 12, 2017, to May 12, 2022)The statistical software’s used for data entry and analysisLine # 35: The authors reported that “The study participants were followed for total of 8237 person-hours of risk time”. Why did the authors employ person-hours of risk? Check it. Person-months will be correct based on the result you report in the entire documentLine #36: the authors reported that “41 (9.9%) of HIV infected children were died”. This value is unclear to me.  Is it cumulative incidence? Or prevalence? Since your data s time to event prevalence is not appropriate estimate. In addition, along with the estimate, you should also report the precision interval.Since estimating time to death is your main objective, we need to know the median survival time with its precision interval. So, I would recommend if you reported it.

**Methods**

Line #141: survival status (death or censored) is not appropriate medical. You can state “Patients who had incomplete data on the outcome variables; like Date of treatment initiation….” Were excluded from the studyLine # 144-151:  The sample size calculation deficit clarification. How the authors compute probability of the event P (E), and number of event (E). I have doubt about the adequacy of the sample size since you are including all the public hospital (8 hospitals)Line # 156: lottery method is not feasible. Simple random sampling methods can be assured by
Lottery methodComputer generated randomizationRandom table 

For your case computer generated randomization will be feasible using your sampling frame

       IV.    Why the authors fail to select some hospitals from the total to make it more feasible 

      V.   Line #159: “Time to death due to HIV/AIDS after initiation of ART” not appropriate; instead replace it with time to death

Line #171: the authors should mention the unit of measurement (hour, day, month, year)Line #173: edit the statement by adding “of” next initiationLine #180-181: the authors mentioned that death unrelated to the outcome of interest were consider as censored. First, how do you confirm the death in unrelated the study outcome? Second, even you the cause of death, treating as censored is not appropriate. There are other remedies like computing risk analysisLine #181: lives longer than the study period is not appropriate description (not accepted ethically), you can use “still on follow up”Line #204: Check the languageThe authors should describe who are the data collectors with their qualificationLine #224: the authors describe “stratified Cox Proportional”, but in the remaining section proportional cox regression model. Should be consistent

**          Result **

Response rate is not appropriate for medical chart review. Use appropriate term. You can say, as per the inclusion criteria, about 97.42% were included for the analysisThe authors should mention the reason of exclusion for the 11 medical chartsLine #246: “resided” checkLine #288-289: the authors mentioned that “The Kaplan-Meier curve also showed thatas the follow-up time was increased, the survival probability of children was decreased”. Is this the correct interpretation of the figure?  because the survival function is always decreasing function over time. So please check your interpretation by considering median survival timeline # 291: “41 (9.9 of the total study participants died)” checkconsider Table 5 as a supplementary file to reduce the number of tables

** Discussions: **

In the discussion section line 357-415, the author compared the findings with previous studies in terms of similarity or difference. However, the underlying causes for the discrepancies between studies need to be elaborated to increase the depth of the discussion and to allow readers know more about the situation.in addition, the authors should mention the public health importance of the finding.

We look forward to receiving your revised manuscript.

Kind regards,

Daniel Bekele Ketema, MPH

Academic Editor

PLOS ONE

Journal Requirements:

<quillbot-extension-portal></quillbot-extension-portal>

---

## [Author Response · Author response to Decision Letter 2]

8 Jun 2023

Manuscript ID: PONE-D-22-31887R2

“Time to death and its predictors among under-five children on antiretroviral treatment in public hospitals of Addis Ababa, Ethiopia, A Retrospective follow up Study."

Thank you for sending us your valuable comments, which immensely improved our manuscript. We included all the editorial comments raised and we also enclosed the point-by-point response of attached here with. It is my pleasure to inform you that the manuscript was edited meticulously.

• A rebuttal letter that responds to each point raised by the academic editor and reviewer(s). We uploaded this letter as a separate file labeled 'Response to Reviewers'.

• A marked-up copy of the manuscript that highlights changes made to the original version. We uploaded this as a separate file labeled 'Revised Manuscript with Track Changes' colored in yellow.

• An unmarked version of revised paper without tracked changes. We uploaded this as a separate file labeled 'Manuscript'.

---

## [Editor Report · Decision Letter 3]

29 Jun 2023

Time to death and its predictors among under-five children on antiretroviral treatment in public hospitals of Addis Ababa, Addis Ababa, Ethiopia, A Retrospective Follow up Study.

PONE-D-22-31887R3

Dear Dr.Belay,

We’re pleased to inform you that your manuscript has been judged scientifically suitable for publication and will be formally accepted for publication once it meets all outstanding technical requirements.

Kind regards,

Daniel Bekele Ketema, MPH

Academic Editor

PLOS ONE

Additional Editor Comments (optional):

Reviewers' comments:

<quillbot-extension-portal></quillbot-extension-portal>

---

## [Editor Report · Acceptance letter]

6 Jul 2023

PONE-D-22-31887R3 

Time to death and its predictors among under-five children on antiretroviral treatment in public hospitals of Addis Ababa, Addis Ababa, Ethiopia, A Retrospective Follow up Study 

Dear Dr. Belay:

I'm pleased to inform you that your manuscript has been deemed suitable for publication in PLOS ONE. Congratulations! Your manuscript is now with our production department. 

Kind regards, 

on behalf of

Daniel Bekele Ketema 

Academic Editor

PLOS ONE